# Normalized Spectral Map Synchronization

**Yanyao Shen**
UT Austin
Austin, TX 78712
shenyanyao@utexas.edu

**Qixing Huang**
TTI Chicago and UT Austin
Austin, TX 78712
huangqx@cs.utexas.edu

**Nathan Srebro**
TTI Chicago
Chicago, IL 60637
nati@ttic.edu

**Sujay Sanghavi**
UT Austin
Austin, TX 78712
sanghavi@mail.utexas.edu

## Abstract

Estimating maps among large collections of objects (e.g., dense correspondences across images and 3D shapes) is a fundamental problem across a wide range of domains. In this paper, we provide theoretical justifications of spectral techniques for the map synchronization problem, i.e., it takes as input a collection of objects and noisy maps estimated between pairs of objects along a connected object graph, and outputs clean maps between all pairs of objects. We show that a simple normalized spectral method (or NormSpecSync) that projects the blocks of the top eigenvectors of a data matrix to the map space, exhibits surprisingly good behavior — NormSpecSync is much more efficient than state-of-the-art convex optimization techniques, yet still admitting similar exact recovery conditions. We demonstrate the usefulness of NormSpecSync on both synthetic and real datasets.

## 1 Introduction

The problem of establishing maps (e.g., point correspondences or transformations) among a collection of objects is connected with a wide range of scientific problems, including fusing partially overlapped range scans [1], multi-view structure from motion [2], re-assembling fractured objects [3], analyzing and organizing geometric data collections [4] as well as DNA sequencing and modeling [5]. A fundamental problem in this domain is the so-called *map synchronization*, which takes as input noisy maps computed between pairs of objects, and utilizes the natural constraint that composite maps along cycles are identity maps to obtain improved maps.

Despite the importance of map synchronization, the algorithmic advancements on this problem remain limited. Earlier works formulate map synchronization as solving combinatorial optimizations [1, 6, 7, 8]. These formulations are restricted to small-scale problems and are susceptible to local minimums. Recent works establish the connection between the cycle-consistency constraint and the low-rank property of the matrix that stores pairwise maps in blocks; they cast map synchronization as low-rank matrix inference [9, 10, 11]. These techniques exhibit improvements on both the theoretical and practical sides. In particular, they admit exact recovery conditions (i.e., on the underlying maps can be recovered from noisy input maps). Yet due to the limitations of convex optimization, all of these methods do not scale well to large-scale datasets.

In contrast to convex optimizations, we demonstrate that spectral techniques work remarkably well for map synchronization. We focus on the problem of synchronizing permutations and introduce a robust and efficient algorithm that consists of two simple steps. The first step computes the top eigenvectors of a data matrix that encodes the input maps, and the second step rounds each block of

the top-eigenvector matrix into a permutation matrix. We show that such a simple algorithm possesses a remarkable denoising ability. In particular, its exact recovery conditions match the state-of-the-art convex optimization techniques. Yet computation-wise, it is much more efficient, and such a property enables us to apply the proposed algorithm on large-scale dataset (e.g., many thousands of objects). Spectral map synchronization has been considered in [12, 13] for input observations between all pairs of objects. In contrast to these techniques, we consider incomplete pair-wise observations, and provide theoretical justifications on a much more practical noise model.

## 2 Algorithm

In this section, we describe the proposed algorithm for permutation synchronization. We begin with the problem setup in Section 2.1. Then we introduce the algorithmic details in Section 2.2.

### 2.1 Problem Setup

Suppose we have $n$ objects $S_1, \cdots, S_n$. Each object is represented by $m$ points (e.g., feature points on images and shapes). We consider bijective maps $\phi_{ij} : S_i \to S_j, 1 \leq i, j \leq n$ between pairs of objects. Following the convention, we encode each such map $\phi_{ij}$ as a permutation matrix $X_{ij} \in \mathcal{P}_m$, where $\mathcal{P}_m$ is the space of permutation matrices of dimension $m$:

$$\mathcal{P}_m := \{X | X \in [0,1]^{m \times m}, X\mathbf{1}_m = \mathbf{1}_m, X^T\mathbf{1}_m = \mathbf{1}_m\},$$

where $\mathbf{1}_m = (1, \cdots, 1)^T \in \mathbb{R}^m$ is the vector whose elements are 1.

The input permutation synchronization consists of noisy permutations $X_{ij}^{\text{in}} \in \mathcal{G}$ along a connected object graph $\mathcal{G}$. As described in [4, 9], a widely used pipeline to generate such input is to 1) establish the object graph $\mathcal{G}$ by connecting each object and similar objects using object descriptors (e.g., HOG [14] for images) , and 2) apply off-the-shelf pair-wise object matching methods to compute the input pair-wise maps (e.g., SIFTFlow [15] for images and BIM [16] for 3D shapes).

The output consists of improved maps between all of objects

$$X_{ij}, 1 \leq i, j \leq n.$$

### 2.2 Algorithm

We begin with defining a data matrix $X^{\text{obs}} \in \mathbb{R}^{nm \times nm}$ that encodes the initial pairwise maps in blocks:

$$X_{ij}^{\text{obs}} = \begin{cases} \frac{1}{\sqrt{d_i d_j}} X_{ij}^{\text{in}}, & (i,j) \in \mathcal{G} \\ 0, & \text{otherwise} \end{cases} \tag{1}$$

where $d_i := |\{S_j | (S_i, S_j) \in \mathcal{G}\}|$ is the degree of object $S_i$ in graph $\mathcal{G}$.

***Remark*** 1. Note that the way we encode the data matrix is different from [12, 13] in the sense that we follow the common strategy for handling irregular graphs and use a normalized data matrix.

The proposed algorithm is motivated from the fact that when the input pair-wise maps are correct, the correct maps between all pairs of objects can be recovered from the leading eigenvectors of $X^{\text{obs}}$:

**Proposition 2.1.** *Suppose there exist latent maps (e.g., the ground-truth maps to one object) $X_i, 1 \leq i \leq n$ so that $X_{ij}^{in} = X_j^T X_i, (i,j) \in \mathcal{G}$. Denote $W \in \mathbb{R}^{nm \times m}$ as the matrix that collects the first $m$ eigenvectors of $X^{\text{obs}}$ in its columns. Then the underlying pair-wise maps can be computed from the corresponding matrix blocks of matrix $WW^T$:*

$$X_j^T X_i = \frac{\sum_{i=1}^n d_i}{\sqrt{d_i d_j}} (WW^T)_{ij}, \quad 1 \leq i, j \leq n. \tag{2}$$

The key insight of the proposed approach is that even when the input maps are noisy (i.e., the blocks of $X^{\text{obs}}$ are corrupted), the leading eigenvectors of $X^{\text{obs}}$ are still stable under these perturbations (we will analyze this stability property in Section 3). This motivates us to design a simple two-step permutation synchronization approach called NormSpecSync. The first step of NormSpecSync computes the leading eigenvectors of $W$; the second step of NormSpecSync rounds the induced

---

**Algorithm 1** NormSpecSync

---

**Input:** $X_{obs}$ based on (1), $\delta_{\max}$

Initialize $W_0$: set $W_0$ as an initial guess for the top-$m$ orthonormal eigenvectors, $k \leftarrow 0$

**while** $\|W^{(k)} - W^{(k-1)}\| > \delta_{\max}$ **do**

    $W^{(k+1)^+} = X^{\mathrm{obs}} \cdot W^{(k)}$,

    $W^{(k+1)} R^{(k+1)} = W^{(k+1)^+}$,     (QR factorization),

    $k \leftarrow k + 1$.

**end while**

Set $W = W^{(k)}$ and $\overline{X}_{i1}^{\mathrm{spec}} = (WW^T)_{i1}$.

Round each $\overline{X}_{i1}^{\mathrm{spec}}$ into the corresponding $X_{i1}$ by solving (3).

**Output:** $X_{ij} = X_{j1}^T X_{i1}$, $1 \le i, j \le n$.

---

matrix blocks (2) into permutations. In the following, we elaborate these two steps and analyze the complexity. Algorithm 1 provides the pseudo-code.

**Leading eigenvector computation.** Since we only need to compute the leading $m$ eigenvectors of $X^{\mathrm{obs}}$, we propose to use generalized power method. This is justified by the observation that usually there exists a gap between $\lambda_m$ and $\lambda_{m+1}$. In fact, when the input pair-wise maps are correct, it is easy to derive that the leading eigenvectors of $X^{\mathrm{obs}}$ are given by:

$$\lambda_1(X^{\mathrm{obs}}) = \cdots = \lambda_m(X^{\mathrm{obs}}) = 1, \ \lambda_{m+1}(X^{\mathrm{obs}}) = \lambda_{n-1}(\mathcal{G}),$$

where $\lambda_{n-1}(\mathcal{G})$ is the second largest eigenvalue of the normalized adjacency matrix of $\mathcal{G}$. As we will see later, the eigen-gap $\lambda_m(X^{\mathrm{obs}}) - \lambda_{m+1}(X^{\mathrm{obs}})$ is still persistent in the presence of corrupted pair-wise maps, due to the stability of eigenvalues under perturbation.

**Projection onto $\mathcal{P}_m$.** Denote $X_{ij}^{\mathrm{spec}} := \frac{\sum_{i=1}^n d_i}{\sqrt{d_i d_j}}(WW^T)_{ij}$. Since the underlying ground-truth maps $X_{ij}, 1 \le i, j \le n$ obey $X_{ij} = X_{jk}^T X_{ik}, 1 \le i, j \le n$ for any fixed $k$, we only need to round $X_{ik}^{\mathrm{spec}}$ into $X_{ik}$. Without losing generality, we set $k = 1$ in this paper.

The rounding is done by solving the following constrained optimization problem, which projects $X_{i1}^{\mathrm{obs}}$ onto the space of permutations via the Frobenius norm:

$$
\begin{aligned}
X_{i1} = \underset{X \in \mathcal{P}_m}{\arg\min} \|X - X_{i1}^{\mathrm{obs}}\|_F^2 &= \underset{X \in \mathcal{P}_m}{\arg\min} \left( \|X\|_F^2 + \|X_{i1}^{\mathrm{obs}}\|_F^2 - 2\langle X, X_{i1}^{\mathrm{obs}}\rangle \right) \\
&= \underset{X \in \mathcal{P}_m}{\arg\max} \ \langle X, X_{i1}^{\mathrm{obs}}\rangle.
\end{aligned}
\tag{3}
$$

The optimization problem described in (3) is the so-called linear assignment problem, which can be solved exactly using the Hungarian algorithm whose complexity is $O(m^3)$ (c.f. [17]). Note that the optimal solution of (3) is invariant under global scaling and shifting of $X_{i1}^{\mathrm{obs}}$, so we omit $\frac{\sum_{i=1}^n d_i}{\sqrt{d_i d_j}}$ and $\frac{1}{m}\mathbf{1}\mathbf{1}^T$ when generating $\overline{X}_{ij}^{\mathrm{obs}}$ (See Algorithm 1).

**Time complexity of NormSpecSync.** Each step of the generalized power method consists of a matrix-vector multiplication and a QR factorization. The complexity of the matrix-vector multiplication, which leverages of the sparsity in $X^{\mathrm{obs}}$, is $O(n_E \cdot m^2)$, where $n_E$ is the number of edges in $\mathcal{G}$. The complexity of each QR factorization is $O(nm^3)$. As we will analyze laser, generalized power method converges linearly, and setting $\delta_{\max} = 1/n$ provides a sufficiently accurate estimation of the leading eigenvectors. So the total time complexity of the Generalized power method is $O\big((n_E m^2 + nm^3)\log(n)\big)$. The time complexity of the rounding step is $O(nm^3)$. In summary, the total complexity of NormSpecSync is $O\big((n_E m^2 + nm^3)\log(n)\big)$. In comparison, the complexity of the SDP formulation [9], even when it is solved using the fast ADMM method (alternating direction of multiplier method), is at least $O(n^3 m^3 n_{admm})$. So NormSpecSync exhibits significant speedups when compared to SDP formulations.

# 3 Analysis

In this section, we provide an analysis of NormSpecSync under a generalized Erdős-Rényi noise model.

## 3.1 Noise Model

The noise model we consider is given by two parameters $m$ and $p$. Specifically, we assume the observation graph $\mathcal{G}$ is fixed. Then independently for each edge $(i, j) \in \mathcal{E}$,

$$X_{ij}^{\text{in}} = \begin{cases} I_m & \text{with probability } p \\ P_{ij} & \text{with probability } 1 - p \end{cases} \tag{4}$$

where $P_{ij} \in \mathcal{P}_m$ is a random permutation.

**Remark** 2. The noise model described above assumes the underlying permutations are identity maps. In fact, one can assume a generalized noise model

$$X_{ij}^{\text{in}} = \begin{cases} X_{j1}^T X_{i1} & \text{with probability } p \\ P_{ij} & \text{with probability } 1 - p \end{cases}$$

where $X_{i1}, 1 \leq i \leq n$ are pre-defined underlying permutations from object $S_i$ to the first object $S_1$. However, since $P_{ij}$ are independent of $X_{i1}$. It turns out the model described above is equivalent to

$$X_{j1} X_{ij}^{\text{in}} X_{i1}^T = \begin{cases} I_m & \text{with probability } p \\ P_{ij} & \text{with probability } 1 - p \end{cases}$$

Where $P_{ij}$ are independent random permutations. This means it is sufficient to consider the model described in (4).

**Remark** 3. The fundamental difference between our model and the one proposed in [11] or the ones used in low-rank matrix recovery [18] is that the observation pattern (i.e., $\mathcal{G}$) is fixed, while in other models it also follows a random model. We argue that our assumption is more practical because the observation graph is constructed by comparing object descriptors and it is dependent on the distribution of the input objects. On the other hand, fixing $\mathcal{G}$ significantly complicates the analysis of NormSpecSync, which is the main contribution of this paper.

## 3.2 Main Theorem

Now we state the main result of the paper.

**Theorem 3.1.** *Let* $d_{\min} := \min_{1 \leq i \leq n} d_i$, $d^{\text{avg}} := \sum_i d_i / n$, *and denote* $\rho$ *as the second top eigenvalue of normalized adjacency matrix of* $\mathcal{G}$. *Assume* $d_{\min} = \Omega(\sqrt{n} \ln^3 n)$, $d^{\text{avg}} = O(d_{\min})$, $\rho < \min\{p, 1/2\}$. *Then under the noise model described above, NormSpecSync recovers the underlying pair-wise maps with high probability if*

$$p > C \cdot \frac{\ln^3 n}{d_{\min}/\sqrt{n}}, \tag{5}$$

*for some constant* $C$.

**Proof Roadmap.** The proof of Theorem 3.1 combines two stability bounds. The first one considers the projection step:

**Proposition 3.1.** *Consider a permutation matrix* $X = (x_{ij}) \in \mathcal{P}_m$ *and another matrix* $\overline{X} = (\overline{x}_{ij}) \in \mathbb{R}^{m \times m}$. *If* $\|X - \overline{X}\| < \frac{1}{2}$, *then*

$$X = \underset{Y \in \mathcal{P}_m}{\arg\min} \|Y - \overline{X}\|_F^2.$$

**Proof.** The proof is quite straight-forward. In fact,

$$\|X - \overline{X}\|_\infty \leq \|X - \overline{X}\| < \frac{1}{2}.$$

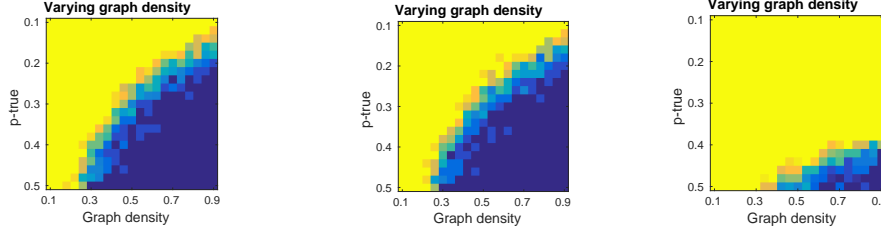

(a) NormSpecSync (2.25 seconds)    (b) SDP (203.12 seconds)    (c) DiffSync(1.07 seconds)

Figure 1: Comparisons between NormSpecSync, SDP[9], DiffSync[13] on the noise model described in Sec. 2.

This means the corresponding element $\overline{x}_{ij}$ of each non-zero element in $x_{ij}$ is dominant in its row and column, i.e.,

$$x_{ij} \neq 0 \ \leftrightarrow \ \overline{x}_{ij} > \max(\max_{k \neq j} \overline{x}_{ik}, \max_{k \neq i} \overline{x}_{kj}),$$

which ends the proof. ∎

The second bound concerns the block-wise stability of the leading eigenvectors of $X^{\mathrm{obs}}$:

**Lemma 3.1.** *Under the assumption of Theorem 3.1, then w.h.p.,*

$$\left\| \frac{\sum_{i=1}^{n} d_i}{\sqrt{d_i d_1}} (WW^T)_{i1} - I_m \right\| < \frac{1}{3}, \quad 1 \leq i \leq n. \tag{6}$$

It is easy to see that we can prove Theorem 3.1 by combing Lemma 3.1 and Prop. 3.1. Yet unlike Prop. 3.1, the proof of Lemma 3.1 is much harder. The major difficulty is that (6) requires controlling each block of the leading eigenvectors, namely, it requires a $L^{\infty}$ bound, whereas most stability results on eigenvectors are based on the $L^2$-norm. Due to space constraint, we defer the proof of Lemma 3.1 to Appendix A and the supplemental material. ∎

**Near-optimality of NormSpecSync.** Theorem 3.1 implies that NormSpecSync is near-optimal with respect to the information theoretical bound described in [19]. In fact, when $\mathcal{G}$ is a clique, (5) becomes $p > C \cdot \frac{\ln^3(n)}{\sqrt{n}}$, which aligns with the lower bound in [19] up to a polylogarithmic factor. Following the model described in [19], we can also assume that the observation graph $\mathcal{G}$ is sampled with a density factor $q$, namely, two objects are connected independently with probability $q$. In this case, it is easy to see that $d_{\min} > O(nq/\ln n)$ w.h.p., and (5) becomes $p > C \cdot \frac{\ln^4 n}{\sqrt{nq}}$. This bound also stays within a polylogarithmic factor from the lower bound in [19], indicating the near-optimality of NormSpecSync.

## 4 Experiments

In this section, we perform quantitative evaluations of NormSpecSync on both synthetic and real examples. Experimental results show that NormSpecSync is superior to state-of-the-art map synchronization methods in the literature. We organize the remainder of this section as follows. In Section 4.1, we evaluate NormSpecSync on synthetic examples. Then in section 4.2, we evaluate NormSpecSync on real examples.

### 4.1 Quantitative Evaluations on Synthetic Examples

We generate synthetic data by following the same procedure described in Section 2. Specifically, each synthetic example is controlled by three parameters $\mathcal{G}$, $m$, and $p$. Here $\mathcal{G}$ specifies the input graph; $m$ describes the size of each permutation matrix; $p$ controls the noise level of the input maps. The input maps follow a generalized Erdos-Renyi model, i.e., independently for each edge $(i,j) \in \mathcal{G}$ in the input graph, with probability $p$ the input map $X_{ij}^{in} = I_m$, and otherwise $X_{ij}^{in}$ is a random permutation. To simplify the discussion, we fix $m = 10, n = 200$ and vary the observation graph $\mathcal{G}$ and $p$ to evaluate NormSpecSync and existing algorithms.

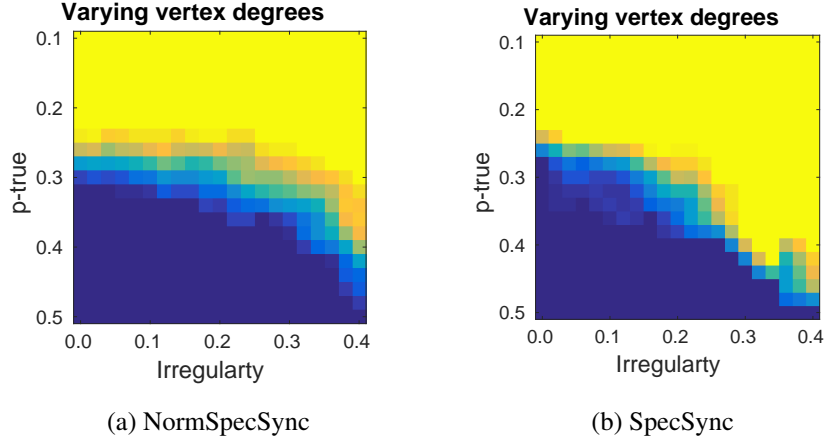

|  (a) NormSpecSync | (b) SpecSync |

Figure 2: Comparison between NorSpecSync and SpecSync on irregular observation graphs.

**Dense graph versus sparse graph.**     We first study the performance of NormSpecSync with respect to the density of the graph. In this experiment, we control the density of $\mathcal{G}$ by following a standard Erdős-Rényi model with parameter $q$, namely independently, each edge is connected with probability $q$. For each pair of fixed $p$ and $q$, we generate 10 examples. We then apply NormSpecSync and count the ratio that the underlying permutations are recovered. Figure 1(a) illustrates the success rate of NormSpecSync on a grid of samples for $p$ and $q$. Blue and yellow colors indicate it succeeded and failed on all the examples, respectively, and the colors in between indicate a mixture of success and failure. We can see that NormSpecSync tolerates more noise when the graph becomes denser. This aligns with our theoretical analysis result.

**NormSpecSync versus SpecSync.**     We also compare NormSpecSync with SpecSync [12], and show the advantage of NormSpecSync on irregular observation graphs. To this end, we generate $\mathcal{G}$ using a different model. Specifically, we let the degree of the vertex to be uniformly distribute between $(\frac{1}{2} - q)n$ and $(\frac{1}{2} + q)n$. As illustrated in Figure 2, when $q$ is small, i.e., all the vertices have similar degrees, the performance of NormSpecSync and SpecSync are similar. When $q$ is large, i.e., $\mathcal{G}$ is irregular, NormSpecSync tend to tolerate more noise than SpecSync. This shows the advantage of utilizing a normalized data matrix.

**NormSpecSync versus DiffSync.**     We proceed to compare NormSpecSync with DiffSync [13], which is a permutation synchronization method based on diffusion distances. NormSpecSync and DiffSync exhibit similar computation efficiency. However, NormSpecSync can tolerate significantly more noise than DiffSync, as illustrated in Figure 1(c).

**NormSpecSync versus SDP.**     Finally, we compare NormSpecSync with SDP [9], which formulates permutation synchronization as solving a semidefinite program. As illustrated in Figure 1(b), the exact recovery ability of NormSpecSync and SDP are similar. This aligns with our theoretical analysis result, which shows the near-optimality of NormSpecSync under the noise model of consideration. Yet computationally, NormSpecSync is much more efficient than SDP. The averaged running time for SpecSync is 2.25 second. In contrast, SDP takes 203.12 seconds in average.

## 4.2   Quantitative Evaluations on Real Examples

In this section, we present quantitative evaluation of NormSpecSync on real datasets.

**CMU Hotel/House.**     We first evaluate NormSpecSync on CMU Hotel and CMU House datasets [20]. The CMU Hotel dataset contains 110 images, where each image has 30 marked feature points. In our experiment, we estimate the initial map between a pair of images using RANSAC [21]. We consider two observation graphs: a clique observation graph $\mathcal{G}_{full}$, where we have initial maps computed between all pairs of images, and a sparse observation graph $\mathcal{G}_{sparse}$. $\mathcal{G}_{sparse}$ is constructed to only connect similar images. In this experiment, we connect an edge between two images if the difference in their HOG descriptors [22] is smaller than $\frac{1}{2}$ of the average descriptor differences among all pairs of images. Note that $\mathcal{G}_{sparse}$ shows high variance in terms of vertex

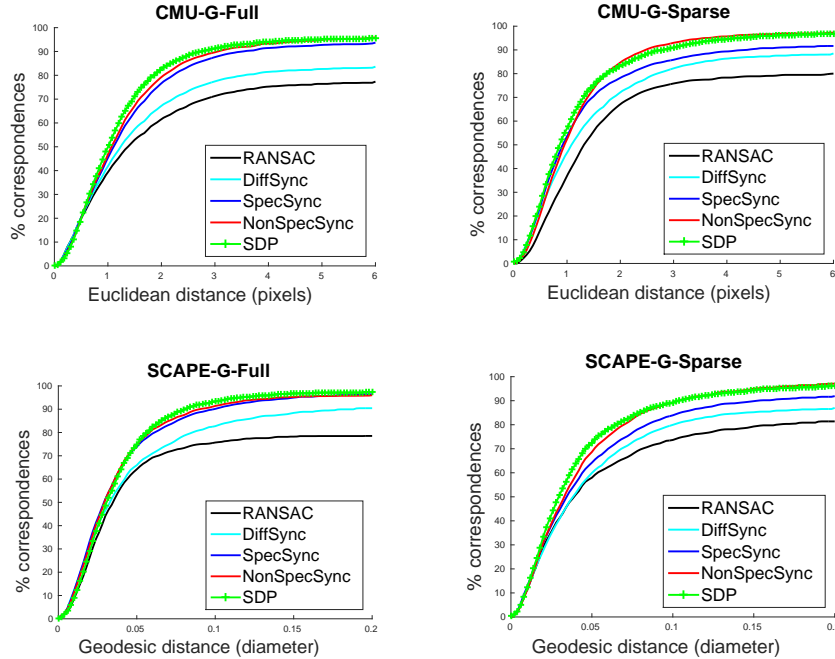

Figure 3: Comparison between NorSpecSync, SpecSync, DiffSync and SDP on CMU Hotel/House and SCAPE. In each dataset, we consider a full observation graph and a sparse observation graph that only connects potentially similar objects.

degree. The CMU House dataset is similar to CMU Hotel, containing 100 images and exhibiting slightly bigger intra-cluster variability than CMU Hotel. We construct the observation graphs and the initial maps in a similar fashion. For quantitative evaluation, we measure the cumulative distribution of distances between the predicted target points and the ground-truth target points.

Figure 3(Left) compares NormSpecSync with the SDP formulation, SpecSync, and DiffSync. On both full and sparse observation graphs, we can see that NormSpecSync, SDP and SpecSync are superior to DiffSync. The performance of NormSpecSync and SpecSync on $\mathcal{G}_{full}$ is similar, while on $\mathcal{G}_{sparse}$, NormSpecSync shows a slight advantage, due to its ability to handle irregular graphs. Moreover, although the performance of NormSpecSync and SDP are similar, SDP is much slower than NormSpecSync. For example, on $G_{sparse}$, SDP took 1002.4 seconds, while NormSpecSync only took 3.4 seconds.

**SCAPE.** Next we evaluate NormSpecSync on the SCAPE dataset. SCAPE consists of 71 different poses of a human subject. We uniformly sample 128 points on each model. Again we consider a full observation graph $\mathcal{G}_{full}$ and a sparse observation graph $\mathcal{G}_{sparse}$. $\mathcal{G}_{sparse}$ is constructed in the same way as above, except we use the shape context descriptor [4] for measuring the similarity between 3D models. In addition, the initial maps are computed from blended-intrinsic-map [16], which is the state-of-the-art technique for computing dense correspondences between organic shapes. For quantitative evaluation, we measure the cumulative distribution of geodesic distances between the predicted target points and the ground-truth target points. As illustrated in Figure 3(Right), the relative performance between NormSpecSync and the other three algorithms is similar to CMU Hotel and CMU House. In particular, NormSpecSync shows an advantage over SpecSync on $\mathcal{G}_{sparse}$. Yet in terms of computational efficiency, NormSpecSync is far better than SDP.

## 5 Conclusions

In this paper, we propose an efficient algorithm named NormSpecSync towards solving the permutation synchronization problem. The algorithm adopts a spectral view of the mapping problem and exhibits surprising behavior both in terms of computation complexity and exact recovery conditions. The theoretical result improves upon existing methods from several aspects, including a fixed obser-

vation graph and a practical noise method. Experimental results demonstrate the usefulness of the proposed approach.

There are multiple opportunities for future research. For example, we would like to extend NormSpecSync to handle the case where input objects only partially overlap with each other. In this scenario, developing and analyzing suitable rounding procedures become subtle. Another example is to extend NormSpecSync for rotation synchronization, e.g., by applying Spectral decomposition and rounding in an iterative manner.

**Acknowledgement.** We would like to thank the anonymous reviewers for detailed comments on how to improve the paper. The authors would like to thank the support of DMS-1700234, CCF-1302435, CCF-1320175, CCF-1564000, CNS-0954059, IIS-1302662, and IIS-1546500.

# A   Proof Architecture of Lemma 3.1

In this section, we provide a roadmap for the proof of Lemma 3.1. The detailed proofs are deferred to the supplemental material.

**Reformulate the observation matrix.**    The normalized adjacency matrix $\bar{A} = D^{-\frac{1}{2}}AD^{-\frac{1}{2}}$ can be decomposed as $\bar{A} = ss^T + V\Lambda V^T$, where the dominant eigenvalue is 1 and corresponding eigenvector is $s$. We reformulate the observation matrix as $\frac{1}{p}M = \bar{A} \otimes I_m + \tilde{N}$, and it is clear to see that the ground truth result relates to the term $(ss^T) \otimes I_m$, while the noise comes from two terms: $(V\Lambda V^T) \otimes I_m$ and $\tilde{N}$. More specifically, the noise not only comes from the randomness of uncertainty of the measurements, but also from the graph structure, and we use $\rho$ to represent the spectral norm of $\Lambda$. When the graph is disconnected or near disconnected, $\rho$ is close to 1 and it is impossible to recover the ground truth.

**Bound the spectral norm of $\tilde{N}$.**    The noise term $\tilde{N}$ consists of random matrices with mean zero in each block. In a complete graph, the spectral norm is bounded by $O(\frac{1}{p\sqrt{n}})$, however, when considering the graph structure, we give a $O(\frac{1}{p\sqrt{d_{\min}}})$ bound.

**Measure the block-wise distance between $U$ and $s \otimes I_m$.**    Let $M = U\Sigma U^T + U_2\Sigma_2 U_2^T$, we want to show the distance between $U$ and $s \otimes \mathbf{1_m}$ is small, where the distance function $dist(\cdot)$ is defined as:
$$dist(U,V) = \min_{R:RR^T=I}\left\|U - VR\right\|_B, \tag{7}$$
and this $B-$norm for any matrix $X$ represented in the form $X = [X_1^T, \cdots, X_n^T]^T \in \mathbb{R}^{mn \times m}$ is defined as
$$\|X\|_B = \max_i \|X_i\|_F. \tag{8}$$
More specifically, we bound the distance between $U$ and $s \otimes I_m$ by constructing a series of matrix $\{A_k\}$, and we can show for some $k = O(\log n)$, the distances from $s \otimes A_k$ to both $U$ and $s \otimes I_m$ are small. Therefore, by using the triangle inequality, we can show that $U$ and $s \otimes I_m$ is close.

**Sketch proof of Lemma 3.1.**    Once we are able to show that there exists some rotation matrix $R$, such that $dist(U, s \otimes I_m)$ is in the order of $o(\frac{1}{\sqrt{n}})$, then it is straightforward to prove Lemma 3.1. Intuitively, this is because the measurements from the eigenvectors is close enough to the ground truth, hence their second moment will still be close. Formally speaking,
$$\left\|U_i U_j^T - (s_i \cdot I_m)(s_j \cdot I_m)\right\| \tag{9}$$
$$=\left\|U_i RR^T U_j^T - (s_i \cdot I_m)(s_j \cdot I_m)\right\| \tag{10}$$
$$=\left\|U_i R(R^T U_j^T - (s_j \cdot I_m)^T) + (U_i R - s_i \cdot I_m)(s_j \cdot I_m)^T\right\| \tag{11}$$
$$\leq\left\|U_i\right\| \cdot dist(U, s \otimes I_m) + dist(U, s \otimes I_m) \cdot \left\|s_j \cdot I_m\right\| \tag{12}$$
On the other hand, notice that
$$\left\|\frac{\sum_{i=1}^n d_i}{\sqrt{d_i d_j}}U_i U_j^T - I_m\right\| = \frac{\sum_{i=1}^n d_i}{\sqrt{d_i d_j}}\left\|U_i U_j^T - (s_i \cdot I_m)(s_j \cdot I_m)\right\|, \tag{13}$$
and we only need to show that (13) is in the order of $o(1)$. The details are included in the supplemental material.

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
