[Supplementary Material · 2486_NormSpecSync_camera_ready_withAppendix.pdf]

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

# A  Reformulating the Noise Term in the Model

As is shown in 3.1, each observed block matrix is corrupted by permutation matrices, therefore the noise can not be directly modeled as random matrices with zero mean, and we need to be careful with it. Since without loss of generality, we can assume each block matrix $X_i$ of the underlying ground truth matrix $X$ to be $I_m$, for convenience, define $X^{\text{gt}}$ as:

$$X^{\text{gt}} = A \otimes I_m + \frac{1-p}{p} A \otimes \left( \frac{1}{m} \mathbf{1_m} \cdot \mathbf{1_m}^T \right). \tag{14}$$

Here, $A$ is the adjacency matrix for graph $\mathcal{G}$. $p$ is the probability that we observe $I_m$ correctly, and with probability $1-p$, the observed block matrix is wrong, i.e., we observe a random permutation. Accordingly, the noise model $N$ we propose written in a block-wise form is:

$$N_{ij} = \begin{cases} \frac{1}{p} I_m - I_m - \frac{1-p}{p} \frac{1}{m} \mathbf{1}_m \cdot \mathbf{1}_m^T, & \text{with probability } p, \\ \frac{1}{p} P_{ij} - I_m - \frac{1-p}{p} \frac{1}{m} \mathbf{1}_m \cdot \mathbf{1}_m^T, & \text{with probability } 1-p, \end{cases} \quad (i,j) \in \mathcal{G}, \tag{15}$$

and for $(i,j) \notin \mathcal{G}$, $N_{ij} = 0$. Intuitively, the bias term $I_m + \frac{1-p}{p} \frac{1}{m} 1 \cdot 1^T$ is added to offset the first moment of the noise matrix and is helpful for our theoretic analysis. The expectation of each $N_{ij}$ is:

$$\mathbb{E}\left[N_{ij}\right] = p\mathbb{E}\left[\frac{1}{p}I_m - I_m - \frac{1-p}{p}\frac{1}{m}\mathbf{1}_m \cdot \mathbf{1}_m^T\right] + (1-p)\mathbb{E}\left[\frac{1}{p}P_{ij} - I_m - \frac{1-p}{p}\frac{1}{m}\mathbf{1}_m \cdot \mathbf{1}_m^T\right] \tag{16}$$

$$= I_m + \frac{1-p}{p}\mathbb{E}\left[P_{ij}\right] - I_m - \frac{1-p}{p}\frac{1}{m}\mathbf{1}_m \cdot \mathbf{1}_m^T \tag{17}$$

$$= 0. \tag{18}$$

We can rewrite our input matrix $X^{\text{in}}$ as a ground truth term with an additive noise term:

$$X^{\text{in}} = pX^{\text{gt}} + pN. \tag{19}$$

Accordingly,

$$X^{\text{obs}} = \left(D^{-\frac{1}{2}} \otimes I_m\right) X^{in} \left(D^{-\frac{1}{2}} \otimes I_m\right). \tag{20}$$

Now, we reconsider the adjacency matrix $A$ of graph $\mathcal{G}$. Ideally, we get a noisy observation of the ground truth matrix $X$ (where all blocks are $I_m$) when graph $\mathcal{G}$ is fully connected. However, in practice, we may encounter problems getting all the pair mapping results, i.e., incomplete observation. Accordingly, only a subset of entries in $A$ is set to be 1, i.e., $A_{ij} = 1$ if $X_{ij}$ is observed, otherwise $A_{ij} = 0$. We denote the normalized adjacency matrix as $\bar{A} = D^{-\frac{1}{2}} A D^{-\frac{1}{2}}$, and we will frequently use the dominant eigenvector $s$ of spectral decomposition of $\bar{A}$:

$$\bar{A} = ss^T + V\Lambda V^T, \tag{21}$$

as well as the second largest eigenvalue of $\bar{A}$, and denote it as $\rho = \|\Lambda\|$.

Furthermore, denote $M^*$ as $pX^{\text{obs}}$, combine previous formula in this section, we have

$$\frac{1}{p}M^* = \bar{A} \otimes I_m + \frac{1-p}{p}\bar{A} \otimes \left(\frac{1}{m}\mathbf{1_m} \cdot \mathbf{1_m}^T\right) + \bar{N} \tag{22}$$

$$= \bar{A} \otimes \left(I_m + \frac{1-p}{p}\frac{1}{m}\mathbf{1_m} \cdot \mathbf{1_m}^T\right) + \bar{N}, \tag{23}$$

where $\bar{N} := \left(D^{-\frac{1}{2}} \otimes I_m\right) N \left(D^{-\frac{1}{2}} \otimes I_m\right)$. We normalize the right hand side of (22) and define $M$ as follows:

$$\frac{1}{p}M \triangleq \left(I_n \otimes \left(I_m - \frac{1-\sqrt{p}}{m}\mathbf{1_m} \cdot \mathbf{1_m}^T\right)\right)\left(\frac{1}{p}M^*\right)\left(I_n \otimes \left(I_m - \frac{1-\sqrt{p}}{m}\mathbf{1_m} \cdot \mathbf{1_m}^T\right)\right) \tag{24}$$

$$= \bar{A} \otimes I_m + \tilde{N}, \tag{25}$$

where

$$\tilde{N} = \left(I_n \otimes \left(I_m - \frac{1-\sqrt{p}}{m}\mathbf{1_m} \cdot \mathbf{1_m}^T\right)\right)\bar{N}\left(I_n \otimes \left(I_m - \frac{1-\sqrt{p}}{m}\mathbf{1_m} \cdot \mathbf{1_m}^T\right)\right). \tag{26}$$

The purpose of our algorithm is to find the spectral decomposition of (22), i.e., the first $m$ eigenvectors of $M^*$, since we can only get $M^*$ from the data but not $M$. However, with basic assumptions on $A$ and $p$ (so that the spectral gap is not too small), we can show that both (22) and (24) give us the same top-m eigenvectors. Therefore, we move our target from (22) to (24) when checking the performance of our algorithm. In the noiseless setting, the algorithm's first $m$ eigenvectors would be $s \otimes I_m$ up to a rotation operation, and we would like to show the convergence to this result in the noisy setting, with reasonable assumption on the value of $p$, $\rho$ and $\|\tilde{N}\|$.

# B  Consistency of Eigenvectors

$\rho, \|\tilde{N}\|$ are the spectral norm of $\Lambda, \tilde{N}$, respectively. We assume

$$\rho + \|\tilde{N}\| < (1+\rho)/2, \qquad \frac{1}{p}\rho + \|\bar{N}\| < 1. \tag{27}$$

We can interpret $\rho$ as the noise from the adjacency graph, and $\tilde{N}$ as the noise from observation. Obviously, $\rho < 1$, and accordingly we have $\rho + \|\tilde{N}\| < 1$, which is a necessary condition for the eigengap. The second condition in (27) implies that $\rho < p$, which means that the graph needs to be well connected enough to tolerate large observation noise. Intuitively, this constraint is due to the interaction between graph noise and observation noise, which is also necessary.

## B.1  Lemma B.1

**Lemma B.1.** *With assumption (27), the top-m eigenvectors of $M$ and $M^*$ are consistent, i.e., they span the same space.*

*Proof.* It is obvious to find the following relationship of $M^*$ and $M$:

$$\frac{1}{p}M = \frac{1}{p}M^*(I_n \otimes (I_m - \frac{1-p}{m}\mathbf{1_m}\cdot\mathbf{1_m}^T)). \tag{28}$$

Hence,

$$\frac{1}{p}(M^* - M) = \frac{1-p}{m}\frac{1}{p}M^*\left(I_n \otimes \mathbf{1_m}\cdot\mathbf{1_m}^T\right) \tag{29}$$

$$= \frac{1-p}{m}\left(\bar{A}\otimes\left(I_m + \frac{1-p}{p}\frac{1}{m}\mathbf{1_m}\cdot\mathbf{1_m}^T\right)\right)\left(I_n\otimes\mathbf{1_m}\cdot\mathbf{1_m}^T\right) \tag{30}$$

$$= \frac{1-p}{p}\left(ss^T + V\Lambda V^T\right)\otimes\left(\frac{1}{m}\mathbf{1_m}\cdot\mathbf{1_m}^T\right) \tag{31}$$

$$= \frac{1-p}{p}\left(\frac{1}{\sqrt{m}}s\otimes\mathbf{1_m}\cdot\frac{1}{\sqrt{m}}s^T\otimes\mathbf{1_m}^T + \sum_{i=1}^{n-1}\sigma_i(\Lambda)\frac{1}{\sqrt{m}}v_i\otimes\mathbf{1_m}\cdot\frac{1}{\sqrt{m}}v_i^T\otimes\mathbf{1_m}^T\right), \tag{32}$$

where (30) is because of the noise term always has a block row (column) sum of 0. In (31) we use the property of Kronecker product, i.e., $(X_1\otimes X_2)(X_3\otimes X_4) = (X_1 X_3)\otimes(X_2 X_4)$, as well as the eigenvector decomposition of $\bar{A}$ defined in (21). Meanwhile, let $v_i$ be the $i$th column of $V$,

$$\frac{1}{p}M^*(v_i\otimes\mathbf{1_m}) = \bar{A}\otimes\left(I_m + \frac{1-p}{p}\frac{1}{m}\mathbf{1_m}\cdot\mathbf{1_m}^T\right)(v_i\otimes\mathbf{1_m}) + \bar{N}(v_i\otimes\mathbf{1_m}) \tag{33}$$

$$= (\bar{A}v_i)\otimes\left(\left(I_m + \frac{1-p}{p}\frac{1}{m}\mathbf{1_m}\cdot\mathbf{1_m}^T\right)\mathbf{1_m}\right) \tag{34}$$

$$= \frac{1}{p}\sigma_i(\Lambda)(v_i\otimes\mathbf{1_m}) \tag{35}$$

Therefore, the difference between $M$ and $M^*$ only includes the change in eigenvalues corresponding to eigenvectors $\frac{1}{\sqrt{m}}v_i\otimes\mathbf{1_m}$ and $\frac{1}{\sqrt{m}}s\otimes\mathbf{1_m}$. On the other hand, since $\rho + \|\tilde{N}\| < 1$, the top-$m$ eigenvalues of $\bar{A}\otimes I_m$ are $[1,1,\cdots,1]$, $\sigma_m(M) > 1 - \|\tilde{N}\| > \rho$, hence the top-$m$ eigenvectors of $M$ do not include $v_i\otimes\mathbf{1_m}$. Meanwhile, since the maximum eigenvalue corresponding to $v_i\otimes\mathbf{1_m}$ for (22) is $\frac{1}{p}\rho$, with assumption (27), the top-$m$ eigenvectors of $M^*$ also do not include $v_i\otimes\mathbf{1_m}$. Therefore, the top-$m$ eigenvectors of $M$ and $M^*$ are consistent.

In this lemma, we try to convey the message that $M^*$ is a perturbation of $M$ that still guarantees the same top-$m$ eigenvector subspace if assumption (27) is satisfied. Moreover, due to the structure of the problem, the perturbation in this problem can be well explained as a summation of $n$ terms, where each of them is a rank-1 matrix spanned by one of the eigenvectors, thus making our analysis easier. ∎

# C  Convergence Analysis

**Lemma C.1.** *Alg.1 has linear convergence rate. More specifically, let $M = U\Sigma U^T + U_2\Sigma_2 U_2^T$, and $diag(\Sigma) = [\sigma_1,\cdots,\sigma_m]$, $diag(\Sigma_2) = [\sigma_{m+1},\cdots,\sigma_{mn}]$, the columns of $U$ are the $m$-dominant eigenvectors, then $\hat{X} = W_i W_i^T$ in Alg.1 converges to $UU^T$ linearly.*

*Proof.* Let $W_k = UC_k + U_2C_{k,2}$, where $U \in \mathbb{R}^{mn \times m}, C_k \in \mathbb{R}^{m \times m}, U_2 \in \mathbb{R}^{mn \times m(n-1)}, C_{k,2} \in \mathbb{R}^{m(n-1) \times m}$. In the following, we assume that $C_k$ is invertible. First of all,

$$W_k^+ = \left(U\Sigma U^T + U_2\Sigma_2 U_2^T\right)(UC_k + U_2 C_{k,2}) = U\Sigma C_k + U_2 \Sigma_2 C_{k,2}. \tag{36}$$

Besides, $C_k, C_{k,2}$ should satisfy the following:

$$I_m = W_k^T W_k = C_k^T C_k + C_{k,2}^T C_{k,2}. \tag{37}$$

Also, since the orthogonality of column vectors of $U$ and $U_2$,

$$U^T W_k^+ = \Sigma C_k, \qquad U_2^T W_k^+ = \Sigma_2 C_{k,2}. \tag{38}$$

According to Alg. 1, let the SVD decomposition of $W_k^+$ be $U_k \Sigma_k^+ V_k^T$, then $U_k$ is what the QR factorization step gives us. Furthermore,

$$W_{k+1}W_{k+1}^T = W_k^+ V_k \Sigma_k^{+^{-2}} V_k^T W_k^{+^T} = W_k^+ \left(W_k^{+^T}W_k^+\right)^{-1} W_k^{+^T}. \tag{39}$$

Substitute (39) with (36), we have:

$$W_{k+1}W_{k+1}^T = W_k^+ \left(W_k^{+^T}W_k^+\right)^{-1} W_k^{+^T} \tag{40}$$

$$= (U\Sigma C_k + U_2\Sigma_2 C_{k,2})\left(C_k^T\Sigma^2 C_k + C_{k,2}^T\Sigma_2^2 C_{k,2}\right)^{-1}(U\Sigma C_k + U_2\Sigma_2 C_{k,2})^T \tag{41}$$

$$= (U\Sigma + U_2\Sigma_2 C_{k,3})\left(\Sigma^2 + C_{k,3}^T\Sigma_2^2 C_{k,3}\right)^{-1}(U\Sigma + U_2\Sigma_2 C_{k,3})^T, \tag{42}$$

where $C_{k,3} = C_{k,2}C_k^{-1}$.

**Proposition C.1.** *Let the minimum eigenvalue in $\Sigma$ be $\sigma_m$, and the maximum eigenvalue in $\Sigma_2$ be $\sigma_{m+1}$, $\sigma_m > \sigma_{m+1}$, we have:*

$$\left\|\Sigma^{-1}X^T\Sigma_2^2 X\Sigma^{-1}\right\|_F \le \frac{\sigma_{m+1}^2}{\sigma_m^2}\left\|X^T X\right\|_F. \tag{43}$$

*This is obvious once we check the eigenvalues on both sides.*

Multiply both sides of (37) with $C_k^{-1}$, we have:

$$C_k^{-1} = C_k^T + C_{k,2}^T C_{k,2}C_k^{-1}, \tag{44}$$

$$\Rightarrow \quad I = C_k C_k^T + C_k C_{k,2}^T C_{k,2}C_k^{-1} = C_k C_k^T \left(I + C_{k,3}^T C_{k,3}\right), \tag{45}$$

$$\Rightarrow C_{k,3}^T C_{k,3} = \left(C_k C_k^T\right)^{-1} - I. \tag{46}$$

According to (40), we have:

$$U^T W_{k+1}W_{k+1}^T U = C_{k+1}C_{k+1}^T = \Sigma \left(\Sigma^2 + C_{k,3}^T\Sigma_2^2 C_{k,3}\right)^{-1}\Sigma^T. \tag{47}$$

Therefore,

$$\left(C_{k+1}C_{k+1}^T\right)^{-1} = \Sigma^{-1}\left(\Sigma^2 + C_{k,3}^T\Sigma_2^2 C_{k,3}\right)\Sigma^{-1} = I + \Sigma^{-1}C_{k,3}^T\Sigma_2^2 C_{k,3}\Sigma^{-1}. \tag{48}$$

According to Proposition C.1 and (46), we have:

$$\left\|(C_{k+1}C_{k+1}^T)^{-1} - I\right\|_F \tag{49}$$

$$= \left\|\Sigma^{-1}C_{k,3}^T\Sigma_2^2 C_{k,3}\Sigma^{-1}\right\|_F \tag{50}$$

$$\le \frac{\sigma_{m+1}^2}{\sigma_m^2}\left\|C_{k,3}^T C_{k,3}\right\|_F \tag{51}$$

$$= \frac{\sigma_{m+1}^2}{\sigma_m^2}\left\|\left(C_k C_k^T\right)^{-1} - I\right\|_F. \tag{52}$$

Therefore, $W_k$ has a linear convergence rate. ∎

# D  Prove Lemma 3.1

## D.1  Lemma D.1

**Lemma D.1.** *The spectral norm of $\bar{N}$ is $O(\frac{\sqrt{\ln n}}{p\sqrt{d_{\min}}})$ with high probability.*

*Proof.* First we introduce the inequality that is useful in the following content, i.e., the Bernstein inequality in the matrix form.

**Proposition D.1. Matrix Bernstein.** *Let $\{S_k\}$ be a serious of independent matrices with same dimension $d_1 \times d_2$, satisfying $\mathbb{E}[S_k] = \mathbf{0}, \|S_k\| \leq L$. Let $Z$ be the sum of matrices $S_k$, i.e., $Z = \sum_k S_k$, and let the "variance" of $Z$ be defined as follow:*

$$v(Z) = max \left\{ \left\| \sum_k \mathbb{E}\left[S_k S_k^*\right] \right\|, \left\| \sum_k \mathbb{E}\left[S_k^* S_k\right] \right\| \right\}. \tag{53}$$

*Then, we have:*

$$\mathbb{E}\left[\|Z\|\right] \leq \sqrt{2v(Z)\log(d_1 + d_2)} + \frac{1}{3}L\log(d_1 + d_2) \tag{54}$$

$$\mathbb{P}(\|Z\| \geq t) \leq (d_1 + d_2)\exp\left(\frac{-t^2/2}{v(Z) + Lt/3}\right), \tag{55}$$

*for $t \geq 0$.*

The $\tilde{N}$ we care about can be written as a summation of $2n$ terms as follows:

$$\tilde{N} = \left(\tilde{N}_{:1} + \tilde{N}_{:2} + \cdots \tilde{N}_{:n}\right) + \left(\tilde{N}_{:1} + \tilde{N}_{:2} + \cdots \tilde{N}_{:n}\right)^T, \tag{56}$$

and $\tilde{N}_{:i} = \mathcal{M}_i(\tilde{N})$, where $\mathcal{M}_i(\cdot)$ operator is a mask operation on a square matrix such that only the first $i$ blocks on the $i$th column are revealed. It is easy to verify that (56) is correct since the diagonal blocks of $\tilde{N}$ are all zero matrices. The norm of $\tilde{N}$ is bounded by two times the norm of the sum of first $n$ terms on the right hand side of (56). Since the $n$ terms are independent, we can use Bernstein inequality.

We observe that,

$$\mathbb{E}\left[\tilde{N}_{:i}\right] = 0, \qquad \|\tilde{N}_{:i}\| \leq \frac{1+p}{p}\frac{1}{\sqrt{d_i d_{\min}}}, \tag{57}$$

$$\left\| \sum_i \mathbb{E}\left[N_{:i}N_{:i}^T\right] \right\| = \left\| \sum_i \mathbb{E}\left[N_{:i}^T N_{:i}\right] \right\| \leq \max_i \left\{ \frac{1}{d_i} \sum_{j \in N(i)} \frac{1}{d_j} \right\} \frac{1-p^2}{p^2} \leq \frac{1}{d_{\min}}\frac{1-p^2}{p^2}. \tag{58}$$

Therefore, as long as $d_{\min} = \Omega(\ln n)$, (55) is dominant by the variance term. Accordingly, $\|\tilde{N}\| = O\left(\frac{\sqrt{\ln n}}{p\sqrt{d_{\min}}}\right)$, w.h.p..

∎

## D.2  Lemma D.2

**Lemma D.2.** *Introduce matrix series $A_{k,i}$, $0 \leq i \leq k$, which are recursively defined as*

$$A_{1,1} = I_m, A_{1,0} = I_m, \tag{59}$$

$$A_{k,0} = \sum_{i=0}^{k-1} A_{k-1,i}B_i, A_{k,i} = A_{k-1,i-1}, 1 \leq i \leq k. \tag{60}$$

*where $B_i = \left(s^T \otimes I_m\right)E^i\left(s \otimes I_m\right)$. Then,*

$$U_k = \sum_{i=0}^{k} E^i\left(s \otimes A_{k,i}\right). \tag{61}$$

*Also,*

$$A_{k+1,0} = U_k^T\left(s \otimes I_m\right) = \left(s^T \otimes I_m\right)M^k\left(s \otimes I_m\right) \tag{62}$$

*Proof.* The proof is quite straightforward and use mathematical induction. When $k = 1$, (61) is true because

$$\sum_{i=0}^{1} E^i (s \otimes A_{1,i}) = E (s \otimes I_m) + (s \otimes I_m) = \left(E + ss^T\right)(s \otimes I_m) = M (s \otimes I_m) = U_1. \quad (63)$$

Suppose (61) is true when $k \leq j - 1, j \geq 2$. Then

$$U_j = \left(\left(ss^T\right) \otimes I_m + E\right) \sum_{i=0}^{j-1} E^i (s \otimes A_{j-1,i}) \quad (64)$$

$$= \sum_{i=0}^{j-1} \left(\left(ss^T\right) \otimes I_m + E\right) E^i (s \otimes A_{j-1,i}) \quad (65)$$

$$= \sum_{i=0}^{j-1} \left(E^{i+1} (s \otimes A_{j-1,i}) + (s \otimes I_m) \left(s^T \otimes I_m\right) E^i (s \otimes I_m) A_{j-1,i}\right) \quad (66)$$

$$= \sum_{i=1}^{j} E^i (s \otimes A_{j-1,i}) + s \otimes A_{j,0} \quad (67)$$

$$= \sum_{i=0}^{j} E^i (s \otimes A_{j-1,i}). \quad (68)$$

∎

## D.3   Lemma D.3

**Lemma D.3. $A_{k,i}$ is close to $I_m$.** *For all $k \leq 10 \log(n)$, we have*

$$\|A_{k,i} - I_m\| \leq \frac{3(k-i)\sqrt{\ln n}}{1 - (\rho + \|\tilde{N}\|)}\|\tilde{N}\|^2, \quad 0 \leq i \leq k. \quad (69)$$

*Proof.* We prove this by using mathematical induction. When $k = 1$, (69) is trivial because the left side is 0. Now, suppose (69) is true for $k \leq j$, and let us consider $k = j + 1$. It is clear that we only need to show that $A_{j+1,0}$ satisfies (69), as the rest of $A_{j+1,i}$ follow directly according to definition. Notice that $\|\tilde{N}\| = O(\frac{\sqrt{\ln n}}{p\sqrt{d_{\min}}})$, it follows that we can choose sufficient large $n$ such that

$$\frac{3k\sqrt{\ln n}}{1 - (\rho + \|\tilde{N}\|)}\|\tilde{N}\|^2 \leq \frac{1}{2}, \quad 1 \leq k \leq 10 \log(n). \quad (70)$$

For $A_{j+1,0}$, we bound the difference between $A_{j+1,0}$ and $I_m$ as follows:

$$\|A_{j+1,0} - I_m\| \leq \|A_{j,0} - I_m\| + \sum_{i=1}^{j} \|A_{j,i}\|\|B_i\| \quad (71)$$

$$\leq \frac{3j\sqrt{\ln n}}{1 - (\rho + \|\tilde{N}\|)}\|\tilde{N}\|^2 + \sum_{i=1}^{j}(1 + \frac{3(j-i)\sqrt{\ln n}}{1 - (\rho + \|\tilde{N}\|)}\|\tilde{N}\|^2)\|B_i\| \quad (72)$$

$$\leq \frac{3j\sqrt{\ln n}}{1 - (\rho + \|\tilde{N}\|)}\|\tilde{N}\|^2 + (1 + \frac{3(j-1)\sqrt{\ln n}}{1 - (\rho + \|\tilde{N}\|)}\|\tilde{N}\|^2)\|B_1\| + \sum_{i=2}^{j} 2\|B_i\| \quad (73)$$

$$\leq \frac{3j\sqrt{\ln n}}{1 - (\rho + \|\tilde{N}\|)}\|\tilde{N}\|^2 + 2p\sqrt{\ln n}\|\tilde{N}\|^2 + \sum_{i=2}^{j} 2(\|\tilde{N}\| + \rho)^{i-2}\|\tilde{N}\|^2 \quad (74)$$

$$\leq \frac{(3j+3)\sqrt{\ln n}}{1 - (\rho + \|\tilde{N}\|)}\|\tilde{N}\|^2. \quad (75)$$

Hence, $A_{j+1,0}$ also satisfies (69). Therefore, the lemma is proved. ∎

## D.4   Lemma D.4

**Lemma D.4.** *The spectral norm of $B_i$ is upper bounded as follows:*

$$\|B_1\| \leq \frac{c\sqrt{\ln n}}{p\sqrt{\sum d_i}}, w.h.p. \qquad \|B_i\| \leq \left(\|\tilde{N}\| + \rho\right)^{i-2}\|\tilde{N}\|^2, \quad \forall i \geq 2. \quad (76)$$

*Proof.* In our problem, according to the definition of $N_{ij}$, it is easy to find that

$$\mathbb{E}[N_{ij} N_{ij}^T] = \frac{1-p^2}{p^2} \left( I_m - \frac{1}{m} \mathbf{1_m} \cdot \mathbf{1_m}^T \right) \tag{77}$$

Therefore, if we let $k$ be the number of matrices (in the form of $N_{ij}$) in the summation $Z$, then, plug in the result to (54) and (55), we have:

$$v(Z) = k \cdot \frac{1-p^2}{p^2}, \quad L = \frac{1+p}{p}. \tag{78}$$

Now, if we let $Z = \sum_{(i,j) \in \mathcal{Z}} N_{ij}$ and $k = |\mathcal{Z}|$. Then,

$$\mathbb{E}[\|Z\|] = \sqrt{2k \frac{1-p^2}{p^2} \log(2m)} + \frac{1+p}{3p} \log(2m), \tag{79}$$

$$\mathbb{P}(\|Z\| \geq t) \leq 2m \cdot \exp(\frac{-t^2/2}{k \cdot \frac{1-p^2}{p^2} + \frac{(1+p)t}{3p}}) \leq \delta. \tag{80}$$

Here, we let the right hand side of the above equation (80) to be less than some value $\delta$, then we have the following restriction for $t$ when solving the inequality on the right:

$$t \geq -\frac{1+p}{3p} \ln(\frac{\delta}{2m}) + \frac{1}{p} \sqrt{\frac{(1+p)^2}{9} \ln^2(\frac{\delta}{2m}) - 2k(1-p^2) \ln(\frac{\delta}{2m})}. \tag{81}$$

If we let $\delta$ to be in the scale of $\frac{1}{k}$, then it is not hard to find that the restriction of $t$ in (81) (in order to have a high probability guarantee on $\|Z\|$) is in $O(\sqrt{k} \ln n)$.

According to definition,

$$B_1 = (s^T \otimes I_m) E(s \otimes I_m) = (s^T \otimes I_m)(V \Lambda V^T \otimes I_m + \tilde{N})(s \otimes I_m) = (s^T \otimes I_m)\tilde{N}(s \otimes I_m). \tag{82}$$

Therefore, the spectral norm of $B_1$ satisfies:

$$\|B_1\| = \|(s^T \otimes I_m)\tilde{N}(s \otimes I_m)\| \tag{83}$$

$$= \| \sum_{(i,j) \in [n] \times [n]} \tilde{N}_{ij} s_i s_j \| \tag{84}$$

$$\leq \| \sum_{(i,j) \in [n] \times [n]} \bar{N}_{ij} s_i s_j \| \tag{85}$$

$$\leq \frac{1}{\sum d_i} \| \sum_{(i,j) \in [n] \times [n]} N_{ij} \| \tag{86}$$

$$\leq \frac{c_1 \sqrt{\ln n}}{p\sqrt{\sum d_i}} \quad w.h.p. \tag{87}$$

$$\tag{88}$$

On the other hand, the proof of the bound for $\|B_i\|$ is as follows:

$$\|B_i\| \leq \|(s^T \otimes I_m)E\| \|E\|^{i-2} \|E(s \otimes I_m)\| \tag{89}$$

$$\leq (\|\tilde{N}\| + \|\Lambda\|)^{i-2} \|\tilde{N}(s \otimes I_m)\|^2 \tag{90}$$

$$\leq (\|\tilde{N}\| + \rho)^{i-2} \|\tilde{N}\|^2. \tag{91}$$

∎

## D.5 Lemma D.5

**Lemma D.5.** *For each fixed $k \geq 2$,*

$$\|E^k(s \otimes I_m)\|_B \leq \frac{ck\sqrt{m}\sqrt{\ln n}}{pd_{\min}} \tag{92}$$

*Proof.* Now, we proceed to bound the term $\|E^k(s \otimes I_m)\|_B$, and we start with simple case when $k$ is small (and just consider spectral norm, since there is only a $\sqrt{m}$ difference). After deriving the bound for small $k$

values, we will use recursion to bound $k$ in general.

$k = 1$. Then,

$$\|(e_1^T \otimes I_m)E(s \otimes I_m)\| \tag{93}$$

$$=\|(e_1^T \otimes I_m)\tilde{N}(s \otimes I_m)\| \tag{94}$$

$$=\|\sum_i \tilde{N}_{1i}s_i\| \tag{95}$$

$$=\frac{s_1}{d_1}\|\sum_i N_{1i}\| \tag{96}$$

$$\leq \frac{c\sqrt{\ln n}}{p\sqrt{\sum d_i}} \tag{97}$$

$$\leq \frac{c\sqrt{\ln n}}{pd_{\min}}. \tag{98}$$

Assume that we have a result in the form of the following:

$$\|(e_1^T \otimes I_m)E^k(s \otimes I_m)\| \leq C_A(k)\sqrt{\ln n} \tag{99}$$

**Proposition D.2.**

$$\mathbb{P}[\|\sum_{i,j}(V\Lambda V^T)_{1i}^t \tilde{N}_{ij}s_j\| > \frac{c\rho^t\sqrt{\ln n}}{p\sqrt{\sum d_i}}] = O(\frac{1}{n}). \tag{100}$$

*This is easy to check using Bernstein inequality and the property that $\sum_i (V\Lambda V^T)_{1i}^{2t} \leq \rho^{2t}$.*

Let's consider general $k = O(\log n)$. Denote $E_{2:n,2:n}$ as the block matrix that removes the first row block and column block from $E$, denote $E_{11}^i$ as the block submatrix of $E^i$ at position $(1,1)$ (instead of $E_{11}$ to the $i$-th power). Also, let $e_{j-1,n-1}$ be the $(n-1) \times 1$ vector where the $(j-1)$-th entry is 1 and all other entries are 0. We have

$$\|(e_1^T \otimes I_m)E^k(s \otimes I_m)\|_F \tag{101}$$

$$=\|(e_1^T \otimes I_m)E^{k-1}\tilde{N}(s \otimes I_m)\|_F \tag{102}$$

$$=\|(E_{11}^{k-1})(e_1^T \otimes I_m)\tilde{N}(s \otimes I_m) + \sum_{i=1}^{k-1}(E_{11}^{k-1-i})\sum_{j=2}^n E_{1j}((e_{j-1,n-1}^T \otimes I_m)E_{2:n,2:n}^{i-1}\tilde{N}_{2:n}(s \otimes I_m))\|_F \tag{103}$$

$$\leq \|E_{11}^{k-1}\|_F\|(e_1^T \otimes I_m)\tilde{N}(s \otimes I_m)\| + \sum_{i=1}^{k-1}\|(E_{11}^{k-1-i})\|_F\|\sum_{j=2}^n E_{1j}((e_{j-1,n-1}^T \otimes I_m)E_{2:n,2:n}^{i-1}\tilde{N}_{2:n}(s \otimes I_m))\|. \tag{104}$$

$$\leq \sqrt{m}(\rho + \|\tilde{N}\|)^{k-1}\|(e_1^T \otimes I_m)\tilde{N}(s \otimes I_m)\| \tag{105}$$

$$+ \sum_{i=1}^{k-1}\sqrt{m}(\rho + \|\tilde{N}\|)^{k-1-i}\|\sum_{j=2}^n \tilde{N}_{1j}((e_{j-1,n-1}^T \otimes I_m)E_{2:n,2:n}^{i-1}\tilde{N}_{2:n}(s \otimes I_m))\| \tag{106}$$

$$+ \sum_{i=1}^{k-1}\sqrt{m}(\rho + \|\tilde{N}\|)^{k-1-i}\|\sum_{j=2}^n (V\Lambda V^T)_{1j}((e_{j-1,n-1}^T \otimes I_m)E_{2:n,2:n}^{i-1}\tilde{N}_{2:n}(s \otimes I_m))\|. \tag{107}$$

In the above equations, we have used the following properties:

$$\|E_{11}^i\|_F^2 = \sum_{j=1}^m \sigma_j^2(E_{11}^i) \leq m \cdot \|E_{11}^i\|^2 \leq m \cdot \|E^i\|^2 \leq m(\rho + \|\tilde{N}\|)^{2i}, \tag{108}$$

To simplify the above equation (106), we define:

$$\mathcal{N}(i,\phi,k) \triangleq \sum_{j \in \phi} \tilde{N}_{kj}((e_j^T \otimes I_m)E_{\phi,\phi}^{i-1}\tilde{N}_\phi(s \otimes I_m)) \tag{109}$$

$$= \sum_{j \in \phi} \tilde{N}_{kj}((e_j^T \otimes I_m)E_{\phi,\phi}^{i-1}\tilde{N}_{\phi,\phi}(s_\phi \otimes I_m)) + \sum_{j \in \phi} \tilde{N}_{kj}((e_j^T \otimes I_m)E_{\phi,\phi}^{i-1}\tilde{N}_{\phi,\bar{\phi}}(s_{\bar{\phi}} \otimes I_m)), \tag{110}$$

where $\phi$ is a subset of $[n]$ and $n - |\phi| \leq k = O(\log n)$.

For the first term in (110), we treat the part $((e_j^T \otimes I_m) E_{\phi,\phi}^{i-1} \tilde{N}_\phi (s_\phi \otimes I_m)$ as fixed matrices, then $\mathcal{N}(i, \phi, k)$ is a sum of independent random variables. Using matrix Bernstein inequality, we have

$$v(\mathcal{N}(i, \phi, k)) = \|\mathbb{E}[\sum_{j \in \phi}(s_\phi^T \otimes I_m)\tilde{N}_\phi E_\phi^{i-1}(e_j \otimes I_m)\tilde{N}_{kj}^T \tilde{N}_{kj}(e_j^T \otimes I_m)E_\phi^{i-1}\tilde{N}_\phi(s_\phi \otimes I_m)]\| \tag{111}$$

$$= \|\sum_{j \in \phi}\frac{1-p^2}{p^2}\frac{1}{d_k d_j}(s_\phi^T \otimes I_m)\tilde{N}_\phi E_\phi^{i-1}(e_j \otimes I_m)(e_j^T \otimes I_m)E_\phi^{i-1}\tilde{N}_\phi(s_\phi \otimes I_m)\| \tag{112}$$

$$\leq \frac{1}{p^2}\|\tilde{N}\|^2(\rho + \|\tilde{N}\|)^{2i-2}\frac{1}{d_k d_{\min}} \tag{113}$$

and each term is bounded by $\frac{1+p}{p}\frac{1}{\sqrt{d_k d_j}}C_A(i)\sqrt{\ln n}$.

For the second term in (110), since $\bar{\phi}$ includes $O(\ln n)$ terms, we can separately bound the parts in it:

$$\|\sum_{j \in \phi}\tilde{N}_{kj}((e_j^T \otimes I_m)E_{\phi,\phi}^{i-1}\tilde{N}_{\phi,\bar{\phi}}(s_{\bar{\phi}} \otimes I_m))\| \tag{114}$$

$$= \|\sum_{j,l \in \phi, r \in \bar{\phi}}\tilde{N}_{kj}((e_j^T \otimes I_m)E_{\phi,\phi}^{i-1}(e_l \otimes I_m)\tilde{N}_{lr}(s_r \otimes I_m))\| \tag{115}$$

$$\leq \|\sum_{j \in \phi}\tilde{N}_{kj}\| \cdot (\rho + \|\tilde{N}\|)^{i-1} \cdot \|\sum_{l \in \phi, r \in \bar{\phi}}\tilde{N}_{lr}s_r\| \tag{116}$$

$$\leq \|\tilde{N}\|(\rho + \|\tilde{N}\|)^{i-1}\frac{c\sqrt{\ln n}}{pd_{\min}} \tag{117}$$

Therefore, combining the bounds for the first and second term in (110) using (113), (117):

$$\|\mathcal{N}(i, \phi, k)\| \leq \frac{1}{p}(\rho + \|\tilde{N}\|)^{i-1}\|\tilde{N}\|\frac{c\sqrt{\ln n}}{d_{\min}}, \tag{118}$$

On the other hand, in order to simplify (107), define:

$$\Lambda(i, 1, [n]/1) \triangleq \sum_{j \in [n]/1}(V\Lambda V^T)_{1j}(e_j^T \otimes I_m)E_{2:n,2:n}^{i-1}\tilde{N}_{2:n}(s \otimes I_m) \tag{119}$$

$$= \sum_{j \in [n]/1}(V\Lambda V^T)_{1j}[(E_{2:n}^{i-1})_{jj}(e_j^T \otimes I_m)\tilde{N}_{2:n}(s \otimes I_m) \tag{120}$$

$$+ \sum_{i'=1}^{i-1}(E_{2:n}^{i-1-i'})_{jj}\sum_{j'=2,j' \neq j}^{n}E_{jj'}(e_{j'}^T \otimes I_m)E_{2:n/j',2:n/j'}^{i'-1}\tilde{N}_{2:n/j'}(s \otimes I_m)], \tag{121}$$

where the second parameter 1 represents the number of $(V\Lambda V^T)$ terms in the expansion. Accordingly,

$$\|\Lambda(i, 1, [n]/1)\| \tag{122}$$

$$\leq \|(E_{2:n}^{i-1})_{jj}\| \cdot \|\sum_{j=2}^{n}\sum_{k \neq j}(V\Lambda V^T)_{1j}\tilde{N}_{jk}s_k\| \tag{123}$$

$$+ \sum_{i'=1}^{i-1}\|(E_{2:n}^{i-1-i'})_{jj}\| \cdot \|\sum_{j=2}^{n}\sum_{j'}(V\Lambda V^T)_{1j}\tilde{N}_{jj'}(e_{j'}^T \otimes I_m)E_{2:n/j'}^{i'-1}\tilde{N}_{2:n/j'}(s \otimes I_m)\| \tag{124}$$

$$+ \sum_{i'=1}^{i-1}\|(E_{2:n}^{i-1-i'})_{jj}\| \cdot \|\sum_{j=2}^{n}\sum_{j'}(V\Lambda V^T)_{1j}(V\Lambda V^T)_{jj'}(e_{j'}^T \otimes I_m)E_{2:n/j'}^{i'-1}\tilde{N}_{2:n/j'}(s \otimes I_m)\| \tag{125}$$

Notice that for (124), we can separate it into two terms to maintain the independence of $\tilde{N}_{jj'}$:

$$\|\sum_{j=2}^{n}\sum_{j'}(V\Lambda V^T)_{1j}^t\tilde{N}_{jj'}(e_{j'}^T \otimes I_m)E_{2:n/j'}^{i'-1}\tilde{N}_{2:n/j'}(s \otimes I_m)\| \tag{126}$$

$$\leq \|\sum_{j=2}^{n}\sum_{j'>j}(V\Lambda V^T)_{1j}^t\tilde{N}_{jj'}(e_{j'}^T \otimes I_m)E_{2:n/j'}^{i'-1}\tilde{N}_{2:n/j'}(s \otimes I_m)\| \tag{127}$$

$$+ \|\sum_{j=2}^{n}\sum_{j'<j}(V\Lambda V^T)_{1j}^t\tilde{N}_{jj'}(e_{j'}^T \otimes I_m)E_{2:n/j'}^{i'-1}\tilde{N}_{2:n/j'}(s \otimes I_m)\| \tag{128}$$

For each term above, we can treat it as a sum of independent random variables. Using similar arguments for bounding $\mathcal{N}$, we get the following bound:

$$\|\sum_{j=2}^{n}\sum_{j'}(V\Lambda V^T)_{1j}^t \tilde{N}_{jj'}(e_{j'}^T \otimes I_m)E_{2:n/j'}^{i'-1}\tilde{N}_{2:n/j'}(s_{2:n/j'} \otimes I_m)\| \tag{129}$$

$$\leq \frac{\rho^t}{p}(\rho + \|\tilde{N}\|)^{i'-1}\|\tilde{N}\|\frac{c\sqrt{\ln n}}{d_{\min}}. \tag{130}$$

Recursively, we have:

$$\|\Lambda(i,t,\phi)\| \leq (\rho + \|\tilde{N}\|)^{i-1}\frac{c\rho^t\sqrt{\ln n}}{p\sqrt{\sum d_i}} \tag{131}$$

$$+ \sum_{i'=1}^{i-1}(\rho + \|\tilde{N}\|)^{i-1-i'}\frac{\rho^t}{p}(\rho + \tilde{N})^{i'-1}\|\tilde{N}\|\frac{c\sqrt{\ln n}}{d_{\min}} \tag{132}$$

$$+ \sum_{i'=1}^{i-1}(\rho + \|\tilde{N}\|)^{i-1-i'} \cdot \|\Lambda(i',t+1,\phi/j')\|. \tag{133}$$

This gives the bound for $\Lambda$:

$$\|\Lambda(k,t,\phi)\| \leq (2\rho + 2\|\tilde{N}\|)^{k-1}\frac{c\rho^t\sqrt{\ln n}}{pd_{\min}}. \tag{134}$$

On the other hand,

$$C_A(k) \leq (\rho + \|\tilde{N}\|)^{k-1}C_A(1) \tag{135}$$

$$+ \sum_{i=1}^{k-1}(\rho + \|\tilde{N}\|)^{k-1-i}(\rho + \|\tilde{N}\|)^{i-1}\frac{\sqrt{\ln n}\|\tilde{N}\|}{pd_{\min}} \tag{136}$$

$$+ \sum_{i=1}^{k-1}(\rho + \|\tilde{N}\|)^{k-1-i}\|\Lambda(i,1,\phi)\|/\sqrt{\ln n}. \tag{137}$$

In the base case when $k=1$, we have:

$$C_A(1) \leq \frac{c}{pd_{\min}}. \tag{138}$$

After recursion steps based on (137), we have:

$$\|E^k(s \otimes I_m)\|_B \leq k(2\rho + 2\|\tilde{N}\|)^{k-1}\frac{c\sqrt{m}\sqrt{\ln n}}{pd_{\min}} \leq \frac{ck\sqrt{m}\sqrt{\ln n}}{pd_{\min}} \tag{139}$$

with high probability, where we require $\rho + \|\tilde{N}\| \leq 1/2$.

∎

### D.6   Lemma D.6

As we already know, $M = U\Sigma U^T + U_2\Sigma_2 U_2^T$. Now, define $F_k = \Sigma^k U^T(s \otimes I_m)$, we have:

**Lemma D.6.** *Let $F_k$ be defined as above. Then, for $k \geq \frac{2}{\log(\frac{2}{1+\rho})}$:*

$$\|U_2\Sigma_2^k U_2^T\| \leq \frac{1}{n^2}, \qquad \|F_k^T F_k - I_m\| \leq \frac{(6k+4)\sqrt{\ln n}}{1-(\rho+\|\tilde{N}\|)}\|\tilde{N}\|^2. \tag{140}$$

*Proof.* According to the eigenvalue stability and assumption (27), we have the following:

$$\sigma_{m+1} < \rho + \|\bar{N}\| < (1+\rho)/2. \tag{141}$$

Taking $k = \frac{2}{\log(\frac{2}{1+\rho})}\log n = O(\log n)$,

$$\|U_2\Sigma_2^k U_2^T\| = \sigma_{m+1}^k(M) \leq \frac{1}{n^2}. \tag{142}$$

Based on the bound of the noises, we will show that $F_k$ is close to $I_m$.

$$\|F_k^T F_k - I_m\| \tag{143}$$

$$=\|F_k^T U^T U F_k - I_m\| \tag{144}$$

$$=\|(M^k(s \otimes I_m) - U_2 \Sigma_2^k U_2^T(s \otimes I_m))^T (M^k(s \otimes I_m) - U_2 \Sigma_2^k U_2^T(s \otimes I_m)) - I_m\| \tag{145}$$

$$\leq\|(s^T \otimes I_m)M^{2k}(s \otimes I_m) - I_m\| + 2\|U_2\Sigma_2^k U_2^T(s \otimes I_m)\|\|U_k\| + \|U_2\Sigma_2^k U_2^T(s \otimes I_m)\|^2 \tag{146}$$

$$\leq\|A_{2k+1,0} - I_m\| + \frac{3}{n^2} \qquad (\text{ since } \|U_k\| \leq \|A_{2k+1,0}\|^{\frac{1}{2}}) \tag{147}$$

$$\leq\frac{(6k+4)\sqrt{\ln n}}{1 - (\rho + \|\tilde{N}\|)}\|\tilde{N}\|^2. \tag{148}$$

∎

The spectral norm bound described above indicates the singular values of $F_k$ fall in

$$\left[1 - \frac{(3k+2)\sqrt{\ln n}}{1 - (\rho + \|\tilde{N}\|)}\|\tilde{N}\|, 1 + \frac{(3k+2)\sqrt{\ln n}}{1 - (\rho + \|\tilde{N}\|)}\|\tilde{N}\|\right].$$

This means that the singular values of $F_k^{-1}$ fall in

$$\left[1 - \frac{4k\ln n}{1 - (\rho + \|\tilde{N}\|)}\|\tilde{N}\|, 1 + \frac{4k\ln n}{1 - (\rho + \|\tilde{N}\|)}\|\tilde{N}\|\right],$$

(which implicitly assumes that $\frac{\sqrt{\ln n}\|\tilde{N}\|^2}{1-(\rho+\|\tilde{N}\|)} < \frac{k-2}{4k(3k+2)}$ and is true for large $n$). Thus, there exists a rotation matrix $R$, such that

$$\|F_k^{-1} - R\| \leq \frac{4k\sqrt{\ln n}}{1 - (\rho + \|\tilde{N}\|)}\|\tilde{N}\|^2. \tag{149}$$

## D.7 Proof of Lemma 3.1

**Lemma D.7.** *With assumption (27), we can conclude that*

$$dist(U, s \otimes I_m) \leq \frac{c_1\sqrt{\ln^3 n}}{p^2 d_{\min}^{3/2}} + \frac{c_2}{n^2} + \frac{c_3\sqrt{\ln^5 n}}{p\sqrt{nd_{\min}}} = O(\frac{\sqrt{\ln^5 n}}{p\sqrt{nd_{\min}}}). \tag{150}$$

*with high probability, where $c_1$, $c_2$, $c_3$ are global constants.*

*Proof.*

$$dist(U, s \otimes I_m) \leq dist(U, s \otimes A_{k,0}) + dist(s \otimes A_{k,0}, s \otimes I_m). \tag{151}$$

In the following, we will bound the two terms on the right hand side of (151) separately. First, let $R$ be the minimizer in the rotation matrix class described in (149). Then,

$$dist(U, s \otimes A_{k,0}) \tag{152}$$

$$\leq\|U - (s \otimes A_{k,0})R\|_B \tag{153}$$

$$=\|M^k(s \otimes I_m)F_k^{-1} - U_2\Sigma_2^k U_2^T(s \otimes I_m)F_k^{-1} - (s \otimes A_{k,0})R\|_B \tag{154}$$

$$=\|M^k(s \otimes I_m)(F_k^{-1} - R) - U_2\Sigma_2^k U_2^T(s \otimes I_m)F_k^{-1} + (M^k(s \otimes I_m) - (s \otimes A_{k,0}))R\|_B \tag{155}$$

$$\leq\|M^k(s \otimes I_m)\|_B\|F_k^{-1} - R\| + \|U_2\Sigma_2^k U_2^T(s \otimes I_m)\|_B\|F_k^{-1}\| + \|(M^k(s \otimes I_m) - (s \otimes A_{k,0}))R\|_B. \tag{156}$$

We bound each term in (156).

$$\|M^k(s \otimes I_m)\|_B \tag{157}$$

$$\leq\|M^k(s \otimes I_m)\|_F \tag{158}$$

$$\leq\sqrt{m}\|M^k(s \otimes I_m)\| \tag{159}$$

$$=\sqrt{m}\|A_{2k+1,0}\|^{\frac{1}{2}} \tag{160}$$

$$\leq\sqrt{2m}. \tag{161}$$

$$\|U_2 \Sigma_2^k U_2^T (s \otimes I_m)\|_B \leq \frac{\sqrt{m}}{n^{10k \cdot \ln \frac{1}{\rho + \|N\|}}} \leq \frac{\sqrt{m}}{n^2}. \tag{162}$$

$$\|(M^k(s \otimes I_m) - (s \otimes A_{k,0}))R\|_B \tag{163}$$

$$= \|\sum_{i=1}^{k} (E^i(s \otimes I_m)A_{k,i})\|_B \tag{164}$$

$$\leq 2 \sum_{i=1}^{k} \frac{ck\sqrt{m}\sqrt{\ln n}}{pd_{\min}} \tag{165}$$

$$\leq \frac{c\sqrt{m}\sqrt{\ln^5 n}}{pd_{\min}}. \tag{166}$$

Finally, we have

$$\|U - (s \otimes A_{k,0})R\|_B \leq \frac{4k\sqrt{2m}\sqrt{\ln n}}{1 - (\rho + \|\tilde{N}\|)}\|\tilde{N}\|^2 + \frac{2\sqrt{m}}{n^2} + \frac{c\sqrt{m}\sqrt{\ln^5 n}}{pd_{\min}}. \tag{167}$$

On the other hand,

$$dist(s \otimes A_{k,0}, s \otimes I_m) \tag{168}$$

$$\leq \max(s)dist(A_{k,0}, I_m) \tag{169}$$

$$\leq \max(s)\|A_{k,0}, I_m\|_F \tag{170}$$

$$\leq \frac{1}{\sqrt{d_{\min}}}\sqrt{m}\|A_{k,0} - I_m\| \tag{171}$$

$$\leq \frac{1}{\sqrt{d_{\min}}} \frac{3k\sqrt{m}\sqrt{\ln n}}{1 - (\rho + \|\tilde{N}\|)}\|\tilde{N}\|^2. \tag{172}$$

Therefore, combining (167) and (172), $dist(U, s \otimes I_m)$ is bounded by:

$$dist(U, s \otimes I_m) \leq \frac{c_1\sqrt{\ln^3 n}}{p^2 d_{\min}^{3/2}} + \frac{c_2}{n^2} + \frac{c_3\sqrt{\ln^5 n}}{pd_{\min}} = O(\frac{\sqrt{\ln^5 n}}{pd_{\min}}). \tag{173}$$

with high probability, where $c_1$, $c_2$, $c_3$ are some global constants. Therefore, the distance is bounded by $O(\frac{\sqrt{\ln^5 n}}{pd_{\min}})$. ∎

Now, we are able to show that there exists some rotation matrix $R$, such that $dist(U, s \otimes I_m)$ is in the order of $o(\frac{1}{\sqrt{n}})$, then it is straightforward to prove Lemma 3.1. This is because:

$$\|U_i U_j^T - (s_i \cdot I_m)(s_j \cdot I_m)\| \tag{174}$$

$$= \|U_i R R^T U_j^T - (s_i \cdot I_m)(s_j \cdot I_m)\| \tag{175}$$

$$= \|U_i R(R^T U_j^T - (s_j \cdot I_m)^T) + (U_i R - s_i \cdot I_m)(s_j \cdot I_m)^T\| \tag{176}$$

$$\leq \|U_i\| \cdot dist(U, s \otimes I_m) + dist(U, s \otimes I_m) \cdot \|s_j \cdot I_m\| \tag{177}$$

Therefore,

$$\|\frac{\sum_{i=1}^{n} d_i}{\sqrt{d_i d_j}} U_i U_j^T - I_m\| \tag{178}$$

$$\leq dist(U, s \otimes I_m) \max(s_i, s_j) \frac{\sum_{i=1}^{n} d_i}{\sqrt{d_i d_j}} \tag{179}$$

$$\leq dist(U, s \otimes I_m) \frac{\sqrt{n}\sqrt{d^{\text{avg}}}}{\sqrt{d_{\min}}}. \tag{180}$$

If we assume that $d^{\text{avg}} = O(d_{\min})$, for $d_{\min} = \Omega(\sqrt{n} \ln^3 n)$, we need:

$$p \geq C \cdot \frac{(\ln n)^3}{d_{\min}/\sqrt{n}}. \tag{181}$$

Without this assumption on the average degree, we need a stronger assumption on the minimum degree, i.e., $d_{\min} = \Omega((nd^{\text{avg}})^{1/3} \ln^2 n)$, and we need:

$$p \geq C \cdot \frac{(\ln n)^3 (nd^{\text{avg}})^{1/2}}{d_{\min}^{3/2}}. \tag{182}$$