[Reviews · NeurIPS 2016]

Reviewer 1

Summary

This paper concentrates on a normalized spectral technique for solving the joint shape mapping / synchronization problem. It provides a theory towards understanding the performance guarantees of the spectral technique. It was claimed by the authors that the spectral technique achieves nearly equivalent performance as SDP, while enjoying much lower computational complexity. Extensive numerical simulations have been carried out to verify the theoretical findings. Overall speaking, spectral technique is definitely a very interesting and efficient paradigm for this joint mapping problem, and hence an attempt to provide rigorous theoretical justification is a very timely and valuable contribution to this field. One of my concerns is that the key sufficient recovery condition (11) is not very easy to interpret, and does not seem to match the prior results well (but I could be wrong). The authors might want to provide more detailed explanation about (11), ideally specializing it to several concrete examples and then comparing each of them with prior results.

Qualitative Assessment

1. The observation matrix X_obs is approximately low-rank as well without normalization. What is the advantage of "normalized" spectral method compared to "unnormalized" spectral method? 2. For the benefits of those not familiar with this subject, it would be good to provide a simple yet concrete example to explain X_{ij}. 3. Define \otimes (Kronecker product) in Equation (2). 4. 3rd paragraph of Section 2.1: {d_i} should be formally defined. What if some d_i = 0? 5. I think in the initialization stage of Algorithm 1, it suffices to say "compute the first few eigenvectors of M using a block power method" (and perhaps provide some reference for the block power method). There is no need to write out all details for the power method as this is fairly standard. 6. For the projection step (i.e.the sentence after Equation (3)): please either explain in detail how to compute it by solving a block-wise linear assignment or cite an appropriate reference, if any. 7. Last sentence of the "Complexity" paragraph in Page 3: For concreteness, the authors might want to mention that full SVD takes time O(n^3) in order to make it more concrete. 8. Equation (5): P_ij has not been formally defined. 9. The authors might need to state explicitly that {X_ij} are independently generated. 10. It is not completely clear to me what Equation (7) means; please define s, V, Lambda more precisely. 11. The notation seems to be inconsistent from here to there. For instance, In Algorithm 1, it says computing the first n eigenvectors, while in the last paragraph of Section 2.3, it says top-m eigenvectors. Please correct the inconsistency. 12. Condition (11), which is perhaps the most important condition herein, is not easy to interpret at least to me. Can the authors specialize it to some simple case (e.g. when the noise is a uniformly random permutation), and then provide a condition in terms of the parameters p, m, and n rather than rho and || N ||? 13. In fact, I'm not sure I understand (11) correctly. The author says rho is the spectral norm of \overline{A}, which is the normalized adjacency matrix of G. Why can we "interpret rho as the noise from the adjacency graph"? The noise is usually thought of as the variance term; what about the mean component? 14. On the other hand, suppose G is a complete graph, then it seems that the adjacency graph is very close to an all-one matrix, and hence \overline{A} is close to 1/n* ones(n,n), and hence rho is almost 1. In this case, what happens to the second condition of (11) if p is bounded away from 1? In prior results, however, G being a complete graph is actually the easiest case, in the sense it can tolerate the largest possible noise (or smallest possible p which is much smaller than 1). 15. In Lemma 3.2, the last sentence reads strange, as "Show that" sounds like an exercise rather than a result. Also, you might want to define "converges linearly" somewhere, as there could be two different interpretation (i.e. geometric convergence rate vs. a convergence rate of O(1/t) ) depending on whether the readers are from the optimization society or not. 16. Typos and minor things: (1) Abstract: "as random permutation matrix" --> "as a random ..." (2) Abstract: "this algorithm NormSpecSync" --> "the proposed NormSpecSync algorithm" (3) 2nd paragraph: "leads remarkable denoising ability" --> "leads to ..." (4) 2nd paragraph: "then find the top eigenvectors" --> "then finds the top eigenvectors"; "project each block" --> "projects each block" (5) 3rd paragraph: "have been considered" --> "has been considered" (6) Section 2.1: "need not obey the consistency property" --> "needs not ..."

Confidence in this Review

2-Confident (read it all; understood it all reasonably well)


Reviewer 2

Summary

This paper proposes a spectral method for extracting consistent maps from a pairwise collection of correspondences. The basic idea is quite simple, essentially extracting a low-rank approximation of a pairwise map matrix (whose rank ideally should be equal to the width of a single map matrix) and then solving a linear assignment problem to project onto the permutation matrices. The basic algorithm here is unsurprising (the approaches to these synchronization problems tend to be either low-rank eigen-approximation [14] or SDP [10]), but it's nice to see that this paper does theoretical analysis with a reasonable noise model --- rather than adding Gaussian noise to the mapping matrices, they consider what happens if an incorrect permutation is added in. I'm not sure how useful even this analysis is in practice (my bet is that correspondence tools make highly-"regular" or predictable mistakes that reinforce more than this random model captures), but the analysis here is much more useful than previous work as far as I know.

Qualitative Assessment

This paper has a simple idea that was communicated in a relatively clear fashion. Consistent map extraction is a tricky problem, and it's nice to see an algorithm that doesn't require SDP to achieve reasonable error bounds. Localized/small-scale comments: - There are a number of English language issues that need to be repaired in the final draft. I'll note a few in this review, but the authors should consider consulting with an additional reader/editor. - l. 2: practice --> practical - l. 29: find --> finds - l. 30: project --> projects - Sec 2.2: Is it really necessary to explicitly choose an eigenvalue algorithm? It seems to me that you might as well just say "compute top m eigenvectors of M*" and let the implementer choose their preferred algorithm. - l. 72 (and several times later): Is it "unary" or "unitary" ? - l. 81: 1s --> ones - Algorithm 1: Please replace "while not converge" with an implementable convergence criterion. Please put something reasonable (ideally a convergence criterion that aligns with the error bounds you need for linear assignment to succeed) - Section 2.3 could use revision from an expository standpoint. In general, you should define AND THEN use a variable; currently, many variables appear in the mathematics far earlier than their intuitive role is explained (e.g. A, p, X). For example, the first few paragraphs are quite frustrating to read because the variable "p" appears without saying what it is. - Sec 2.3: What is the difference between X and X_gt? This lost me by eq (6) - l. 90: "we observe a random permutation" - Eq (7): Again, what is s? Make sure all the math is self contained and makes sense reading the paper linearly from beginning to end. - I did not check the proofs in sec3 closely but they do appear reasonable. - I'm not sure if readers will find section 3.4 valuable. It seems halfway between two reasonable approaches to this paper --- either just state the final mathematical result with detailed discussion of how to interpret it, or include the entire mathematical proof. Including a few formulas without proof as steps of the larger proof seems less than valuable. - l. 225: It seems NorSpecSync and SpecSync have quite similar behavior in this test. Is there a test where the difference is not clear? Also, please include SpecSync in Figure 1. - For the SCAPE test, my understanding is that there are papers in the graphics world that are specifically dedicated to surface-to-surface map synchronization that may be worth comparing to.

Confidence in this Review

2-Confident (read it all; understood it all reasonably well)


Reviewer 3

Summary

This paper propose simple but effective Normalized Spectral Map Synchronization algorithm to solve map synchronization problem. The proposed method is analysed both theoretically and empirically and shows competitive result with state-of-the-art methods with much less computation complexity.

Qualitative Assessment

The proposed method is simple and practically effective in map synchronization problem. As the performance is guaranteed by theoretical analysis and the the performance and efficiency over other methods is illustrated quantitatively, the proposed method is expected to be applied widely in related problems. The presentation of method, analysis and experiments are clear.

Confidence in this Review

1-Less confident (might not have understood significant parts)


Reviewer 4

Summary

This paper provides theoretical justifications for spectral techniques [14,15] for the map synchronization problem. Compared with state-of-the-art convex optimization techniques e.g. SDP based synchronization method [12], the propose approach is more scalable to thousands of objects, meanwhile admits exact recovery condition similar to other methods like SDP based method [12] in that it extends the Gaussian assumption of the noise to random permutations of the map noise, which allows for incomplete pair-wise observations as the input of the method.

Qualitative Assessment

The evaluation shall involve more state-of-the-art methods for SDP method based study e.g. reference [12] rather than the earlier one [10], also another missing reference shall be compared or discussed while it also considers the node affinity when the synchronization is enforced: @inproceedings{zhou2015multi, title={Multi-image matching via fast alternating minimization}, author={Zhou, Xiaowei and Zhu, Menglong and Daniilidis, Kostas}, booktitle={Proceedings of the IEEE International Conference on Computer Vision}, pages={4032--4040}, year={2015} } The overall novelty of this paper is a little weakened by the reference [12] where the bound and recovery condition is analyzed for the SDP based method case. Some other loosely relevant recent work on pairwise matching synchronization can be found in the following two papers, where the synchronization has another approximate term consistency: [a] Multi-graph matching via affinity optimization with graduated consistency regularization, T-PAMI 2016 [b] Consistency-driven alternating optimization for multigraph matching: A unified approach, T-IP 2015

Confidence in this Review

2-Confident (read it all; understood it all reasonably well)